# Efficacy of Feed-Based Formalin-Killed Vaccine of *Streptococcus iniae* Stimulates the Gut-Associated Lymphoid Tissues and Immune Response of Red Hybrid Tilapia

**DOI:** 10.3390/vaccines9010051

**Published:** 2021-01-14

**Authors:** Mohammad Hayat, Md Sabri Mohd Yusoff, Mohd Jamil Samad, Intan Shameha Abdul Razak, Ina Salwany Md Yasin, Kim D. Thompson, Khalil Hasni

**Affiliations:** 1Department of Veterinary Pathology and Microbiology, Faculty of Veterinary Medicine, Universiti Putra Malaysia, UPM Serdang, Selangor 43400, Malaysia; hayatvet91@gmail.com (M.H.); mjamil@upm.edu.my (M.J.S.); 2Aquatic Animal Health and Therapeutics Laboratory, Institute of Bioscience, Universiti Putra Malaysia, UPM Serdang, Selangor 43400, Malaysia; salwany@upm.edu.my; 3Department of Pre-Clinical Science, Faculty of Veterinary Medicine, Universiti Putra Malaysia, UPM Serdang, Selangor 43400, Malaysia; intanshameha@upm.edu.my; 4Department of Aquaculture, Faculty of Agriculture, Universiti Putra Malaysia, UPM Serdang, Selangor 43400, Malaysia; 5Aquaculture Research Group, Moredun Research Institute, Pentland Science Park, Bush Loan, Penicuik, Midlothian EH26 0PZ, UK; kim.thompson@moredun.ac.uk; 6Faculty of Marine Science, Lasbela University of Agricultural, Water and Marine Science, Uthal District Lasbela, Balochistan 90150, Pakistan; khalil_hasni1@yahoo.com

**Keywords:** oral vaccination, *Streptococcus iniae*, red hybrid tilapia, IgM, gut-associated lymphoid tissue, formalin-killed oral vaccine

## Abstract

Red hybrid tilapia were fed a formalin-killed oral *Streptococcus iniae* vaccine (FKV) in the present study was assessed. Three hundred Red hybrid tilapia 80 ± 10 g were divided into five groups (1A, 1B, 2A, 2B, and Cx), each consisting of 60 fish. Fish from Groups 1A, 1B, 2A, and 2B were fed with FKV over different periods of administration, while Group 2B was the only group of fish to receive an oral booster vaccination on day 14- and 21-days post-vaccination (dpv). Group Cx was fed with normal pellets containing no vaccine as a control group. At four weeks post-vaccination (wpv), all fish were experimentally infected with *S. iniae*. Groups 2A and 2B had the lowest level of mortalities following vaccination (45% and 30%, respectively) compared to Groups 1A and 1B (80% and 55%, respectively), while the level of mortalities in Group Cx was 100%. All vaccinated groups showed a significant increase in anti-*S. iniae* IgM levels (*p* < 0.05) in serum, mucus, and gut-lavage, while Group Cx did not (*p* > 0.05) and all fish in this group died by five weeks post-infection. In conclusion, fish fed with the *S. iniae* FKV had a greater level of protection against *S. iniae*, with increased specific antibody response to the vaccine and there was also evidence of GALT stimulation by the vaccine.

## 1. Introduction

Tilapia, *Oreochromis* sp. is an important and economically significant fish species for aquaculture globally. It is therefore important to improve their resistance to endemic diseases [1]. Streptococcosis, caused either by *Streptococcus agalactiae* or *Streptococcus iniae*, is one of the major bacterial diseases affecting this fish species worldwide [2,3,4,5]. Mortalities in affected fish are associated with septicaemia, meningoencephalitis, loss of orientation, ulcers, lethargy, and exophthalmia [6]. Streptococcosis, caused either by *S. agalactiae* or *S. iniae* leads to levels of high mortality, resulting in severe economic losses for tilapia farmers [6]. Streptococcal septicaemia arising from *S. iniae* infection was reported as a major contributor to low productivity and economic loss in tilapia [7]. *Streptococcus iniae* seems to be expanding its definitive host, with its recent isolation from fish species such as Red Porgy (*Pagrus pagrus*) [8], hybrid tilapia (*Oreochromis* sp.), and freshwater Asian seabass (*Lates calcarifer*) [9]. In Malaysia, outbreaks of streptococcosis caused by *S. iniae*, have been associated with high mortality in Red hybrid tilapia [10].

Vaccination remains the best and most practical way of preventing streptococcosis in fish. Currently, there are various formulations of vaccines used against the disease, delivered mainly through injection administration. Oral vaccination has many benefits, however, including low cost to produce, ease of administration, less stressful for the fish relative to other routes of delivery, and does not require any vaccination equipment [11]. In fish, gut-associated lymphoid tissues; (GALT) acts similarly to the tonsils and Peyer’s patches present in mammals, with the production and recruitment of lymphocytes after antigen stimulation of the mucosa [12]. Mucosal sites within the fish are actively involved in preventing infection from microbes through the action of T and B lymphocytes that reside in the mucosae [13,14], and exposure to antigen through oral vaccination mimics this. Hence, the efficacy of oral vaccines has often been investigated by evaluating the ability of antigens to induce lymphocytes aggregations and immune response in the mucosal regions, including the production of specific antibodies against the vaccine. Several assays for measuring *S. iniae*-specific antibody responses in tilapia have already been described [15,16,17,18,19].

There are recent reports of vaccination being used against streptococcosis in Malaysia tilapia aquaculture [11,20], using both feed-based formalin-killed vaccines (FKV) and feed-based adjuvant vaccines [20]. Nur-Nazifah et al. [21] reported that a feed-based recombinant vaccine of *S. agalactiae* produced significantly higher IgM antibody levels in the mucus, serum, and gut lavage of tilapia compared to fish vaccinated with a FKV. However, no study has been conducted to evaluate the efficacy of a formalin-killed vaccine (FKV) administered orally against streptococcosis caused by *S. iniae*. The objective of the current study was, therefore, to examine the efficacy of an *S. iniae* FKV, delivered orally to Red hybrid tilapia through diet, to evaluate the humoral antibody response elicited by the vaccine in serum, mucus and gut-lavage of vaccinated fish and to assess any stimulation of GALT within the lamina propria by the vaccine.

## 2. Materials and Methods

### 2.1. Fish and Feeding

A total of 310 Red hybrid tilapia (*Oreochromis* sp.) with an average weight of 80 ± 10 g were obtained from Aquaculture Extension Center (AEC), Department of Fisheries, Bukit Tinggi, Pahang, Malaysia, with no previous history of streptococcosis. The fish were acclimatized for 14 days at the Aquatic Animal Health Unit (AAHU), Faculty of Veterinary Medicine, Universiti Putra Malaysia. Ten Red hybrid tilapia were sacrificed to evaluate their health status, screening for bacterial and parasitic infections and to confirm they were *S. iniae*-free. The fish were divided into five groups, each consisting of 60 tilapia, and placed in five 2000 L replicate tanks (10 fish per tank). An automated aerator was used to provide uninterrupted aeration throughout the experiment. The fish were fed a commercial starter feed (Cargill Malaysia (code no: 6243-2M, size 4 mm, protein 47%, fat 10%)) twice daily at 2% of tank biomass. This experiment was approved by the Institutional Animal Care and Use Committee, Universiti Putra Malaysia (IACUC/AUP-R094/2018).

### 2.2. Bacterial Stock

The *Streptococcus iniae* isolates used in this study were obtained from the bacterial collection held at the Institute of Bioscience, Universiti Putra Malaysia, and which has been isolated from Red hybrid tilapia at Kenyir Lake, Terengganu, Malaysia in 2014.

### 2.3. Preparation of Streptococcus iniae for Challenge

The *S. iniae* was subcultured on tryptic soy agar (TSA, Merck, Darmstadt, Germany) and incubated at 30 °C for 48 h. Five bacterial colonies from the TSA plate were further subcultured into 100 mL of the tryptic soy broth (TSB, Merck, Germany) and incubated in a shaker incubator at 30 °C for 48 h. The next day, approximately 0.5 mL of the culture broth was inoculated into a tilapia by intraperitoneal injection. *Streptococcus iniae* was re-isolated from the kidney, eyes, and brain of the fish, which died within 24–48 h after infection. To confirm the identity of the recovered bacteria, API rapid ID 32 Strep^®^ (BioMerieux SA, Marcy I’Etoile, France) was used according to the manufacturer’s instruction and further verified through the polymerase chain reaction (PCR) described below. Thereafter, 10 colonies of the recovered *S. iniae* were subcultured into 100 mL of TSB until it reached a logarithmic growth phase. To determine the bacterial concentration of the suspension, the *S. iniae* culture (1 mL) was added into peptone water (9 mL) and a 10-fold serial dilution prepared (10^1^ to the lowest 10^9^), then 0.1 mL of each serial dilution was streaked onto the TSA plate prior to incubation at 30 °C for 24–48 h. The next day colonies (between 30 and 300) were counted according to Alcamo et al. [22] and the concentration presented as colony-forming units per millilitre (CFU/mL). The final concentration of the live *S. iniae* used for the experimental challenge was 1 × 10^6^ CFU/mL. The latter was subcultured into TSB and incubated using a shaker incubator at 300× *g* and 30 °C for 19 h, to obtain the growing cell. Finally, the desired concentration of 1 × 10^6^ CFU/mL was used for the live *S. iniae* challenge of vaccinated fish.

### 2.4. Preparation of the Inactivated Cells

The *S. iniae* was streaked onto TSA and incubated at 30 °C for 24–48 h and resulting *S. iniae* colonies subcultured into TSB and cultured as described previously. Buffered formalin (0.5%) was added to the bacteria suspension and incubated overnight at 4 °C to inactivate the bacteria. The inactivated bacteria were subsequently harvested and washed with sterile phosphate-buffered saline (PBS) centrifuging at 5000× *g* at 4 °C for 5 min. The wash step was carried out 3 times to ensure removal of the formalin. The inactivated cells were resuspended in sterile PBS and using the McFarland standard. Approximately 1 mL of the vaccine was streaked onto blood agar, to confirm its sterility. The mixture was finally incubated at 37 °C for 24–48 h.

### 2.5. Dietary Vaccine Preparation

The formalin-killed bacteria were resuspended in PBS (1 × 10^6^ CFU/mL in 500 mL) (FKV) and incorporated onto the commercial diet (Cargill, Malaysia) using a sprayer machine to spray evenly over all the diet pellets. The pellets were dried for 48 h at 30 °C prior to vaccinating the fish.

### 2.6. Experimental Design

#### 2.6.1. Vaccination of Fish

Three hundred red hybrid tilapia were divided into five Groups: 1A, 1B, 2A, 2B, and Cx. Each group consisted of 60 tilapia, held in five 2000 L tank replicates per group (10 fish per tank). Group 1A (*n* = 60) was vaccinated continuously for 3 days by feeding the FKV coated diet. Group 1B (*n* = 60), was vaccinated continuously for 6 days by feeding the FKV coated diet. Group 2A (*n* = 60) was also vaccinated for 9 days by feeding the FKV coated diet, while Group 2B (*n* = 60), was fed continuously with the FKV-coated diet for 9 days and then given a booster vaccination with the oral vaccine on 14- and 21-days post-vaccination (dpv). The control Group Cx, (*n* = 60), on the other hand, was not vaccinated and was fed the normal commercial feed (Cargill, Malaysia) throughout the vaccination trial, used as a non-vaccinated control group (further details of the experimental design are shown in Table 1). During vaccination trial, five fish from each group of tilapia were sacrificed weekly for the collection of intestinal samples to investigate the development of GALT, and mucus, gut-lavage fluid, and serum were collected for the evaluation levels of the anti-*S. iniae* IgM to the FKV vaccine. The remaining fish in each group (40 per group) were used for the challenge trial at 4 weeks post-vaccination (wpv).

#### 2.6.2. Experimental Infection of Vaccinated Fish with Streptococcus iniae

At 4-wpv, all experimental groups were experimentally infected with *S. iniae* by intraperitoneal injection. The replicate groups of fish were transferred into five 2000 L tanks (40 per tank) without running water. Tricaine methanesulfonate (MS-222) solution (1 L) was used to anaesthetize the fish, and 0.5 mL of the inoculum described in Section 2.3 at 1 × 10^6^ CFU/mL was intraperitoneally injected into each fish. Physical and behavioral changes associated with streptococcosis and mortality rate were monitored for 14 dpi. Fish displaying clinical signs of disease were euthanized for necropsy as soon as signs appeared. All fish surviving at the end of the challenge trial were killed using MS-222. A post-mortem examination was conducted, and samples collected from lesions in the spleen, stomach, kidney, brain to confirm the presence of the bacterial. The experimental design was approved in compliance with the humane methods recommended by the Institutional Animal Care and Use Committee (IACUC) of Universiti Putra Malaysia (AUP No.: UPM/ACUC/AUP-R094/2018).

#### 2.6.3. Collection of Serum, Mucus and Gut-Lavage Fluid

Blood (500 μL) was collected from each fish through the caudal peduncular vein, transferred into a centrifuge tube and stored at 4 °C for 4 h to allow blood to clot. The sample was centrifuged at 3000 rpm for 3 min, serum collected and stored at −20 °C. Sterile cotton was used to collect the mucus from the skin, wiping the surface multiple times with the same cotton swab before immersing it into 1 mL bijou bottles containing sterile PBS + 0.02% (*w*/*v*) sodium azide (PBSS). The solution was kept at 4 °C overnight and centrifuged as described for the serum and stored at −20 °C prior until analyzed by ELISA. The hindgut was sampled for collection of lavage fluid by selecting 10 cm length of the gut, and immediately immersing it into 1 mL of sterile PBSS. The gut was massaged gently prior to collecting the fluid. The fluid was centrifuged as described for the serum and the mucus to eliminate any debris and approximately 500 μL of the gut-lavage was stored at −20 °C for analysis.

#### 2.6.4. Enzyme-Linked Immunosorbent Assay

An indirect ELISA was conducted with the blood serum, mucus and gut-lavage, sampled before and after vaccination to measure the antibodies levels against *S. iniae* as described by Firdaus et al. [20]. Colonies of *S. iniae* were obtained from the TSA plates, subcultured into 100 mL of the TSB and incubated on a shaker-incubator at 30 °C for 24 h. The culture was washed three times with PBS, centrifuging at 5000× *g* for 15 min on each wash. The concentration of the bacterial suspension was determined as described above and the final bacterial concentration adjusted to 1 × 10^6^ CFU/mL. the bacteria were pelleted and carbonate–bicarbonate coating buffer (pH 9.6) was used to re-suspend the pellet before boiling the bacteria in a water bath at 97 °C for 20 min. Once cool, microtiter ELISA plates were coated in triplicate with the *S. iniae* suspension containing 1 × 10^6^ CFU/100 μL per well and the plates incubated overnight at 4 °C. The ELISA plates were then washed with PBS + 0.05% Tween-20 (PBST). A blocking buffer (PBST containing bovine serum albumin) was used to block non-specific binding. Plates were washed three times before as described before the samples, diluted at 1:1000, were added to the well at 100 μL and incubated at 37 °C for 60 min. The ELISA procedure described by Firdaus et al. [20] was employed to quantify the antibodies level. PBST was used to wash the samples and followed by addition of goat anti-tilapia immunoglobulin serum. The latter was diluted at 1:5000, which was then added into each well and incubated for an additional 60 min. The same procedure was conducted before adding 100 μL of the conjugated rabbit anti-goat IgM-horseradish peroxidase (Nordic, the Netherlands), diluted same as GAT. The same washing steps were carried out, followed by addition of the substrate containing dimethyl sulfoxide (DMSO) (Merck, Rogers, AR, USA), tetramethyl 3,3′,5,5′-benzidine (Merck), 0.1 M sodium acetate/citric acid buffer and 3% hydrogen peroxide into each well. The reaction was incubated for 30 min. To terminate the reaction, 50 μL of 2 M sulphuric acid was added to each well before reading the plates at a wavelength of 450 nm (340st; Anthos Zenyth, Salzburg, Austria).

#### 2.6.5. Bacterial Isolation and Gram Stain

Swabs from kidney, brain, and eyes of sacrificed/dead fish were streaked unto TSA at weekly intervals post-challenge. The medium was incubated at 30 °C for 48 h. Suspected colonies of *S. iniae* were confirmed by the presence of Gram-positive cocci. As well as Gram staining, a commercial test kit, API rapid ID 32 Strep^®^ (bioMerieux SA, Marcy I’Etoile, France) was also applied to characterize the bacterial colonies.

#### 2.6.6. DNA Extraction and PCR

A Wizard Genomic DNA Purification Kit (Promega, Madison, MI, USA) was used to extract DNA from samples. A conventional PCR using S. iniae specific 16S rDNA primers SILOX-1 forward [5′-AAG GGG AAA TCG CAA GTC CC-3′] and SILOX-2 reverse [5′-ATA TCT GAT TGG GCC GTC TAA-3′], (Apical Scientific, Malaysia) was performed [23]. Each 50 µL reaction contained 5 µL genomic DNA; 1 µL both forward and reverse primers (50 pmoles µL^−1^ each), 25 μL MyTaq Red master mix (Bioline, London, UK), and 18 µL of distilled water. Known *S. iniae* positive DNA and deionized water were used as positive and negative controls, respectively. The PCR consisted of an initial denaturation at 95 °C for 1 min, followed by 30 cycles at 95 °C for 15 s, annealing at 52 °C for 1 min, and extension at 72 °C for 30 s using a Mastercycler^®^ pro (Eppendorf, Hamburg, Germany). The amplicons were run in 1.5% agarose gel electrophoresis mixed with GelRed Nucleic Acid Gel Stain (Biotium, Fremont, CA, USA) in 1 h at 100 V, 400 A and viewing in the UV transilluminator.

#### 2.6.7. Histological Analysis

The effect of vaccination on the GALT of vaccinated fish was assessed histologically post-vaccination. The dorsal part of the hindgut was excised and fixed in buffered formalin (10%) for at least 1 day. Alcohol solutions were used to dehydrate the samples for 24 h, before using xylene for clearing. Samples were impregnated with paraffin wax with a melting point of 57 °C. Gut samples were then sectioned at 4 μm and transferred to glass slides, allowed to dry overnight at 40 °C, then stained with haematoxylin and eosin (H & E) stain. A total of 10 microscopic fields of 3 guts from each group were examined randomly, GALT’s diameter and numbers of cells within the GALT were measured using FIVE Image Analyzer (Olympus, Tokyo, Japan).

#### 2.6.8. Statistical Analysis

SPSS software (IBM SPSS Statistics for Windows, Version 27.0. Armonk, NY, USA: IBM Corp) was used to conduct the Statistical analysis (IgM levels, GALT size). A one-way analysis of variance (ANOVA) was used for comparisons of the means and Tukey HSD was employed for checking the significance of the results, *p* < 0.05 considered to be significant differences between groups.

## 3. Results

### 3.1. Antibody Response

#### 3.1.1. Antibody Response Prior to Vaccination

Before the onset of vaccination at zero-week post-vaccination (wpv), levels of *S. iniae* antibody in the gut-lavage fluids, blood serum, and body mucus were very low (Figure 1, Figure 2 and Figure 3).

#### 3.1.2. Serum Antibody Response

Oral immunization of the Red hybrid tilapia with FKV in Groups 2A and 2B resulted in significant (*p* < 0.05) increases in the serum antibody levels (IgM) as early as 1-wpv. However, the level of specific antibodies in the serum of unvaccinated Group Cx did not significantly change (*p* > 0.05) over the vaccination period (Figure 1). The greatest increase in antibody levels was seen in Group 2B at 6-wpv following the first, and subsequent booster vaccinations at 2- and 3-wpv, respectively and subsequently challenge with *S. iniae*. In fact, the antibody levels in all vaccinated groups (1A, 1B, 2A, and 2B) began to increase significantly (*p* < 0.05) until the termination of the trial at 6 wpv. Similar serum IgM levels were obtained for Groups 1A, 1B, and 2A; however, antibody levels in Groups 1A and 1B were lower compared to Group 2A. There was no significant change in the antibody pattern of Group Cx throughout the trial, and all fish died in this group by one-week post-challenge (wpc).

#### 3.1.3. Mucus Antibody Responses

In vaccinated groups 1A, 1B, 2A, and 2B mucus antibody level increased over the course of the trial, which started from 1-wpv. In Groups 2A and 2B the mucus antibody levels increased significantly (*p* < 0.05) due to nine days of vaccine administration in both groups with first and second boosters to Group 2B at weeks 2 and 3 (Figure 2). The mucus IgM levels of Groups 1A, 1B exhibited a similar pattern as those of Group 2A but remained lower than those of Group 2A. As found with the serum, there were no significant changes in the mucus antibody levels in Group Cx and all fish had died in this group by 5-wpv.

#### 3.1.4. Gut-Lavage Antibody Response

The level of specific IgM in the gut-lavage of Groups 1A, 1B, 2A, and 2B all increased over the course of the vaccination, increasing week by week in all groups until the end of the trail at 6-wpv. The gut-lavage fluid antibody levels of Group 2B were significantly (*p* < 0.05) higher followed by Groups 1B and 2A (Figure 3). The IgM levels of Group 1B showed an identical pattern as those of Group 2A but remained than. At post-challenge, all vaccinated groups showed a similar pattern of antibody response as seen with mucus IgM levels, with only a slightly change in the control Group Cx, with no significant difference in gut-lavage antibody levels seen in this group.

### 3.2. Challenge Trial

#### 3.2.1. Clinical Findings, Mortality Rate and Percentage of Survival

The clinical signs typical of a streptococcal infection were apparent in all groups of fish as early as 12 h post-challenge. These included inappetence, unilateral and bilateral exophthalmia, body discoloration, haemorrhagic eye, lethargic and erratic swimming, and were more pronounced in the unvaccinated control Group Cx compared to the vaccinated groups of fish. Necropsy findings of the streptococcal infection in Red hybrid tilapia are presented in Figure 4 and Figure 5. Group 2B had the lowest level of mortality with 70% survival compared to Groups 1A, 1B, 2A, and Cx, which had survival levels of 20%, 45%, 55%, and 0%, respectively (Table 2).

#### 3.2.2. Bacterial Isolation and PCR

Specific mortalities resulting from the *S. iniae* infection was confirmed from the successful re-isolation of the bacterium from the eye, brain and kidney of infected fish, as early as 6 to 48 h post-challenge as shown in Table 3. These isolates were confirmed as *S. iniae* by PCR using SILOX1/SILOX2 primers, which produced a band of 870 bp [24].

#### 3.2.3. Histological Analysis of the Hindgut

The GALT was clearly observed in the histological sections of hindgut sampled from the vaccinated Red hybrid tilapia. Feeding the FKV to the tilapia induced the presence of lymphoid cell aggregations within the lamina propria of all vaccinated groups of fish (Groups 1A, 1B, 2A, and 2B). Lymphocytes were seen distributed in the epithelium of the GALT as early as 1-wpv, as shown in Figure 6a–d), while lymphoid cell aggregation were absent in the lamina propria of Red hybrid tilapia hindgut in unvaccinated control Group Cx as shown in Figure 7.

The diameter of the GALT and the number of lymphocyte cells within GALT were assessed in the vaccinated Red hybrid tilapia. The diameter of GALTs in Groups 1A and 1B were significantly lower (*p* < 0.05) compared to Groups 2A and 2B. Further, lymphocytes number within the GALT in Groups 1A and 1B were significantly lower (*p* < 0.05) compared to groups 2A and 2B, whereas there was no occurrence of lymphoid cell aggregation found in the hindgut of Red hybrid tilapia of unvaccinated control Group Cx

#### 3.2.4. Size of the GALT in Different Vaccination Groups

There was an increase in the GALT diameter one week after vaccination in all vaccination groups (Figure 8). Groups 2A and 2B had the largest diameters diameter (289 μm and 342 μm, respectively), at 6-wpv (2-wpc) and these were significant different (*p* < 0.05) to other groups. The highest GALT diameter recorded for Group 1A and 1B were 205 μm and 222 μm, respectively, at 6-wpv and were significantly lower to the other two vaccination groups at this time (*p* < 0.05).

#### 3.2.5. Number of Lymphoid Cells

As shown in Figure 9, the number of lymphocytes within the GALT sections of Groups 1A, 1B, 2A, 2B increased equally as early as 1-wpv with 198, 195, 199, and 202 lymphocytes counted per sections, respectively. However, by 3-wpv, the number of lymphocytes in the GALT of Groups 2A and 2B GALT had increased significantly (*p* < 0.05) compared to the other groups, and this increased continued until 6-wpv. The highest number of lymphocytes observed within GALT of Groups 1A and 1B were 392 and 433 cells, respectively, at 6-wpv. However, Groups 2A and 2B showed significantly higher numbers of lymphoid cells (*p* < 0.05), by comparison, with 679 and 808 lymphocytes counted, respectively, at 6-wpv.

## 4. Discussion

*Streptococcus iniae* is considered to be one of the most important streptococcal pathogens of cultured fish [5]. This pathogen continues to have a worldwide distribution, based on recent reported outbreaks and isolation of the bacterium from tilapia and other fish species in Brazil [7], Argentina [16], Indonesia [23], Thailand [25] and China [26]. In Malaysia, *S. iniae* was successfully isolated from outbreaks in farms where *S. agalactiae* infections had previously been recorded [27]. Moreover, severe economic losses due to high mortality are associated with these outbreaks, especially during the dry season [28].

Vaccination remains the most effective approach for prevention and control of streptococcal infections in cultured fish [29]. Amongst the different routes of delivery used for vaccine administration, oral delivery is the most practicable method for vaccinating fish within an aquaculture setting, especially if there is a need to vaccinate large numbers of fish within the farm [30,31]. Many different commercial vaccines are available for streptococcosis, but the majority of these are administrated by intraperitoneal injection or by immersion, and are designed for use on for large farms, which have suitable facilities and technical support to perform the labour-intensive vaccine procedure. However, it is unlikely that small scale fish farmers in South East Asia countries, such as Malaysia, Thailand, and Indonesia would use these types of vaccines. This emphasizes the need for easier, cost-effective, and less demanding vaccination methods to encourage these farmers to vaccinate their fish [30,31]. Oral vaccination also has several benefits for large scale producers, including less handling of fish, less stressful for the fish, and lower costs for the manpower and specialized equipment for the vaccination procedure [31].

This study describes the protective capacity of a FKV for streptococcosis in Red hybrid tilapia, delivered by oral vaccination and tested by experimentally infecting fish with *S. iniae* by intraperitoneal injection. The purpose of oral administration of this vaccine is to stimulate both mucosal and humoral immune responses of immunized fish to promote protection against streptococcosis caused by *S*. *iniae*. We also described changes in GALT morphology in the lamina propria of Red hybrid tilapia after vaccination and challenged. The results of the study showed that Red hybrid tilapia vaccinated orally with FKV produced greater antibody levels in serum, gut-lavage, and body mucus compared to non-vaccinated fish. A significant increase in serum, gut-lavage, and body mucus antibody levels has also been previously reported after intraperitoneal challenge in tilapia give a feed-based vaccine or whole-cell biofilm vaccines prior to challenge [30,31,32]. Another study showed that a feed-based recombinant vaccine produced from *S. agalactiae* induced greater IgM levels compared to those immunized with FKV [21]. Such observation reflects the capacity of the vaccine to enhance both mucosal and humoral immunity in vaccinated tilapia, evidenced here by significant increases in IgM levels in serum, gut-lavage, and body mucus, with levels of antibodies increasing after challenge, showing that the *S. iniae* infection acts as an additional immunological boost. The present study was not designed to compare the humoral immune response efficacies between different vaccination routes; Wang et al. [33], for example reported a significant increase in antibody levels following intraperitoneal vaccination with a FKV against *S. iniae* infection in tilapia. An intraperitoneal infection route was used to infect the fish to ensure that a systemic immune response was obtained from the oral vaccination. The efficacies of vaccines are strongly influenced by the routes of vaccination [33]. Systemic immune responses are readily elicited by intraperitoneal and intramuscular injections compared to oral routes and provide better levels of protection [34]. However, both local and systemic immune responses are elicited when a sufficient amount of antigen is transported and reaches the second gut segment of the fish [35]. Teleost fish possess a second gut segment that is pivotal in oral vaccination, antigen is taken up by intraepithelial macrophages and lymphoid tissues as the vaccine is transportation through the intestinal lumen [36]. After antigens are phagocytosed by macrophages, the cells migrate to other lymphoid organs, inducing a systemic immune response [37]. High levels of protection were obtained in vaccinated Red hybrid tilapia compared to the unvaccinated control group. However, the groups that did not receive the booster vaccinations, which had significantly higher levels of survival by comparison. The group receiving the booster vaccination had 70% of fish surviving the *S. iniae* challenge, indicating the importance of giving a booster vaccination to increase and maintain levels of protection. The levels of survival in the other vaccination groups (1A, 1B, and 2A) and the unvaccinated control (Cx) were 20%, 45%, 55%, and 0%, respectively. These results are consistent with the findings of Ismail et al. [29], who observed a significant rise in serum IgM levels in tilapia for up to six weeks following administration of a second booster feed-based vaccine containing *S. agalactiae*. They obtained higher levels of survival in the fish given a double booster for three weeks (70% survival) compared to fish only given one dose of the vaccine (50% survival). Hence, these further support previous studies where repeated vaccine dosage enhances immune response and extended duration of protection [29,38]. With protection levels against *S. iniae* of 70%, our vaccine performed very well, although Chettri et al. [39] suggested that protection levels should be greater than 80% for the vaccine to be considered excellent.

Mucosal tissues are important portals of entry for streptococcal pathogens to gain access to the fish’s systemic circulation [40]. Therefore, vaccines that stimulate both systemic and mucosal immunity, as seen with the oral vaccination in our study, are ideal for controlling *S. iniae* infection. Moreover, immunoglobulins in fish, such as IgT/IgZ, have been reported a vital component of the mucosal immune response. Antigen uptakes at mucosal surfaces have been linked to the role of intestinal T cells, which is equally important for mucosal targeted vaccines [41,42,43].

Many different studies have evaluated the clinical signs and pathogenicity of *S. iniae* in streptococcosis. In our study, the clinical signs observed in the Red hybrid tilapia infected with *S. iniae* included bilateral exophthalmia, body discoloration, haemorrhagic skin, haemorrhagic eye, lethargy, erratic swimming, and anorexia. Notably, these signs were more obvious in the unvaccinated compared to vaccinated groups, with mortality levels of 100% obtained in the unvaccinated group compared to vaccinated fish (below 80%). Rahmatullah et al. [25] reported similar clinical signs in Red hybrid tilapia infected with *S. iniae*, with mortalities ranging from 90 to 100% within 14 days post-infection, while external signs, such as cachexia and exophthalmia, were observed in Blue Nile tilapia (*Oreochromis aureus*) infected with the bacterium [16]

The necropsy of infected fish showed signs of septicaemia, splenomegaly, empty stomach, intracerebral haemorrhage, and haemorrhagic nephritis, all typical of streptococcal infection. These findings were consistent with the observations of Ortega et al. [16] following *S. iniae* infection in two different tilapia populations. A recent study also reported the presence of prominent lesions in the brain, spleen, and kidney in necropsied offshore cage-cultured fish (*Trachinotus ovatus*) and *S. iniae* was identified as the causative agent [16].

PCR is frequently used to detect and diagnose *S. iniae* infections [23,24,25,27,44,45,46]. In one study, primers were designed for the *lactate oxidase* gene of *S. iniae*, using one-step PCR that yielded an 870 bp fragment specific for the bacteria [24]. Rodkhum et al. [25], on the other hand, developed a duplex-PCR for *S. iniae* based on amplification of 16s rRNA gene producing a 220 bp amplicon. In our study, the bacteria were successfully detected using a conventional PCR that amplified an 870 bp region of the 16s rRNA gene [24], confirming that the pathogen isolated from the various organs (brain, kidney and eye) of the challenged fish was *S. iniae*.

GALT has an important role in mucosal immunity of the fish gut; GALT in the lamina propria of fish gut contains macrophages, plasma cells, and lymphocytes [47,48]. The GALT is formed in fish following the accumulation of lymphocytes at the loci where the antigen is presented in the lamina propria [21,32]. The histology performed on the various groups of fish, allowed us to examine the effect of the oral vaccine on GALT morphology, and to vaccine uptake into the gut lumen [46]. We found increased GALT development in the hindgut of vaccinated Red hybrid tilapia as early as 1-wpv in all groups of vaccinated fish, with both the diameter of the GALT and associated lymphocyte populations increasing in vaccinated fish over the course of the experiment, while GALT tissue of unvaccinated control group remained unchanged. Overall, the results are consistent with reports of Firdaus-Nawi et al. [31] who assessed mucosal immunity and protection for a feed-based adjuvant vaccine against *S. agalactiae*-induced streptococcosis in tilapia. Despite the aggregation of GALT in the lamina propria of the gut, the diameters of GALT between vaccinated groups were not significantly different except between those vaccinated once weekly and the group vaccinated five times per week [31]. However, in this study, both the diameter of GALT between the vaccinated groups and between those given booster are significantly different. This further explains the need to consider the frequency of vaccinations for the stimulation of GALTs and its size with respect to FKV against *S. iniae.* In previous studies, the size of the GALT and the lymphocyte population within the GALT correlated with antibody secretion in the gut-lavage fluid [31]. Accordingly, such associations are suggestive of antigen uptake by the intestinal epithelial cells after vaccination, followed by transportation of antigen to antigen-presenting cells (APCs) and stimulation of B and T cells [48]. Nevertheless, such assumption requires more investigation in relation to *S. iniae* and the protective capacity of the FKV delivered orally to tilapia.

## 5. Conclusions

The findings of this study suggest that the FKV vaccine delivered orally to Red hybrid tilapia provides protection against streptococcosis by eliciting a systemic and mucosal immune response against *S. iniae*. Enhanced protection is achieved by administrating a second booster dose of the FKV vaccine to fish, resulting in increased survival of fish and development of GALT with increased in numbers of lymphocytes within the gut lamina propria of vaccinated Red hybrid tilapia.

## Figures and Tables

**Figure 1 vaccines-09-00051-f001:**
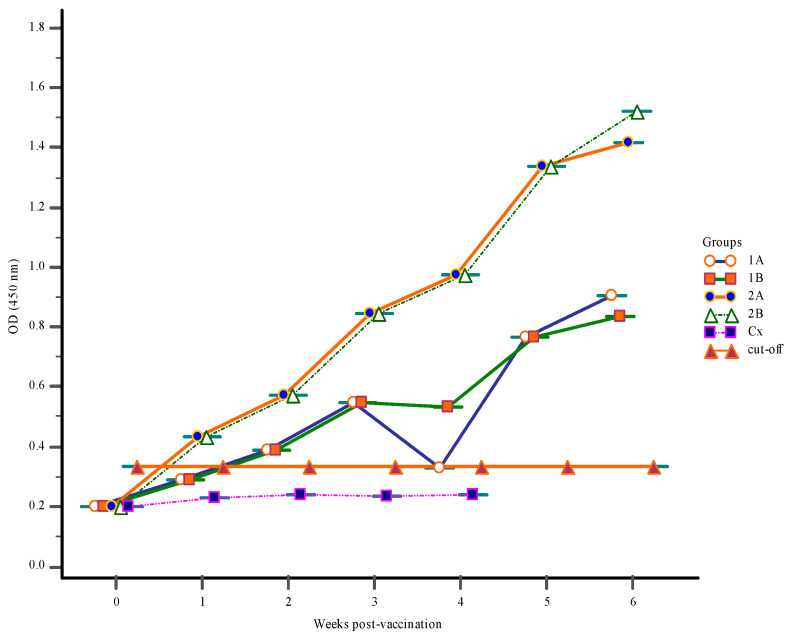
The serum IgM response following vaccination with a feed-based vaccine (1A, 1B, 2A, 2B) and control Group Cx. The first vaccination was done at 0-week post-vaccination (wpv) for groups (1A, 1B, 2A, 2B) with a 1st and 2nd booster vaccination given to Group 2B at 2- and 3-wpv, respectively. All groups were challenged with 10^6^ CFU/mL live *S. iniae* via intra-peritoneum route at 4-wpv.

**Figure 2 vaccines-09-00051-f002:**
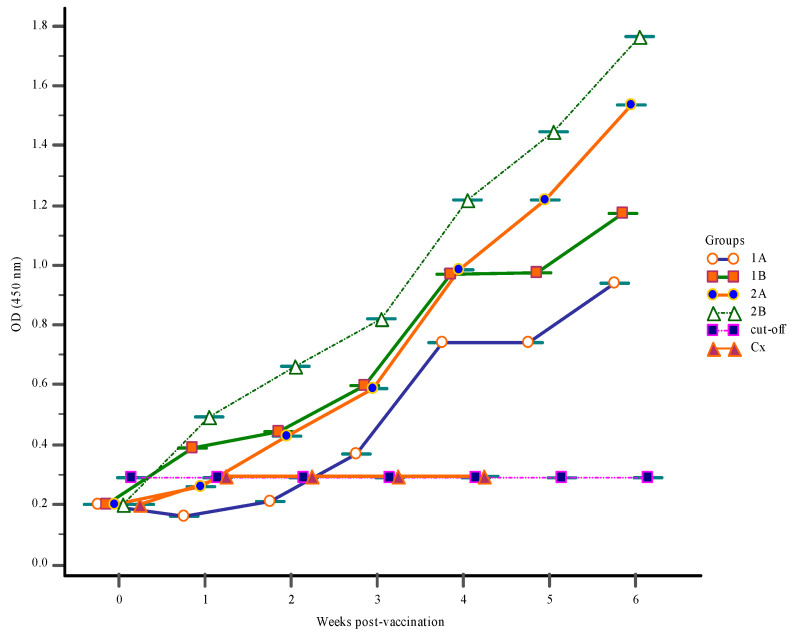
The mucus IgM response following vaccination with feed-based vaccine (1A, 1B, 2A, 2B) and control Group Cx. The first vaccination was done at 0-week post-vaccination (wpv) for groups (1A, 1B, 2A, 2B) with a 1st and 2nd booster vaccination given to Group 2B at 2- and 3-wpv, respectively. All groups were challenged with 10^6^ CFU/mL live *S. iniae* via intra-peritoneum route at 4-wpv.

**Figure 3 vaccines-09-00051-f003:**
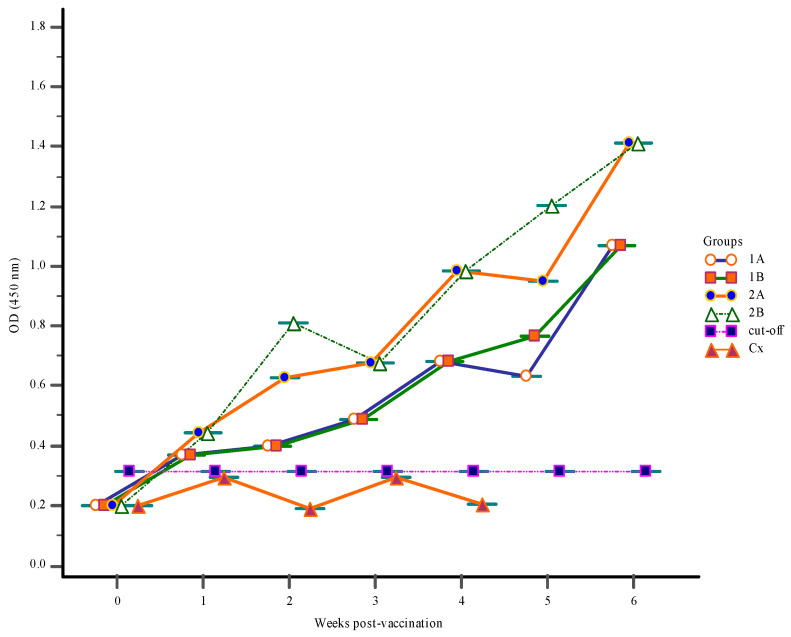
The gut-lavage fluid IgM response following vaccination with feed-based vaccine (1A, 1B, 2A, 2B) and control Group Cx. The first vaccination was done at 0-week post-vaccination (wpv) for groups (1A, 1B, 2A, 2B) with a 1st and 2nd booster vaccination given to Group 2B at 2- and 3-wpv, respectively. All groups were challenged with 10^6^ CFU/mL live *S. iniae* via intra-peritoneum route at 4-wpv.

**Figure 4 vaccines-09-00051-f004:**
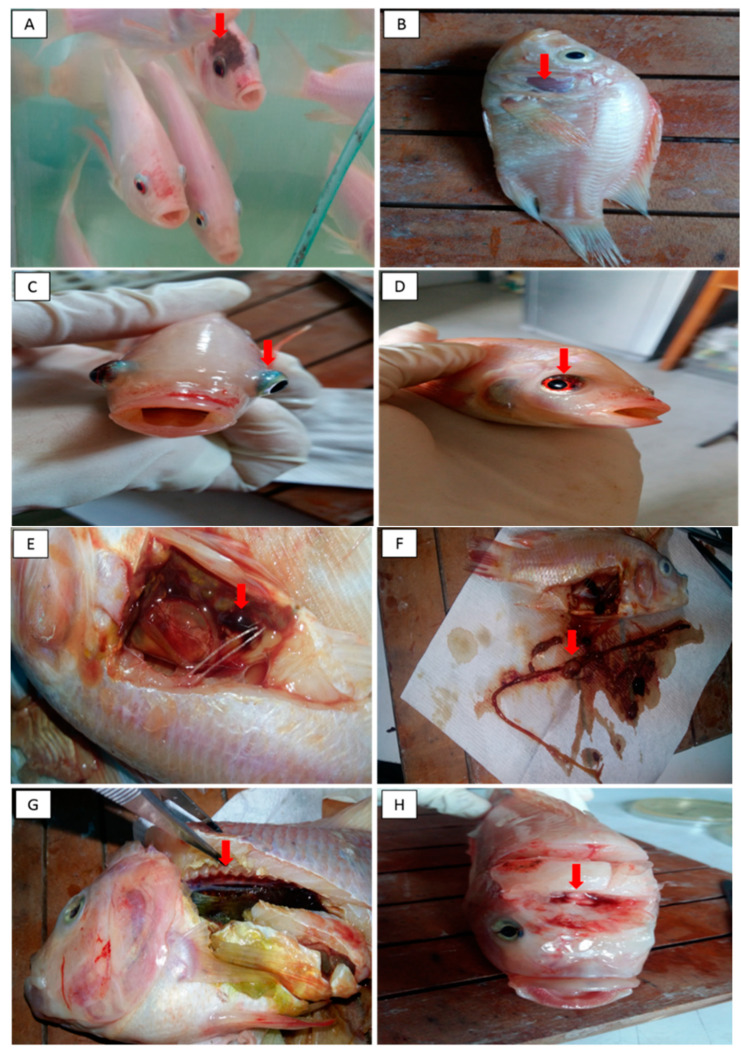
(**A**) skin discoloration, (**B**) haemorrhagic skin, (**C**) exophthalmia (pop-eye), (**D**) haemorrhagic eye (**E**) splenomegaly with multiple granulomas (**F**) empty stomach (**G**) nephritis (**H**) intracerebral haemorrhage.

**Figure 5 vaccines-09-00051-f005:**
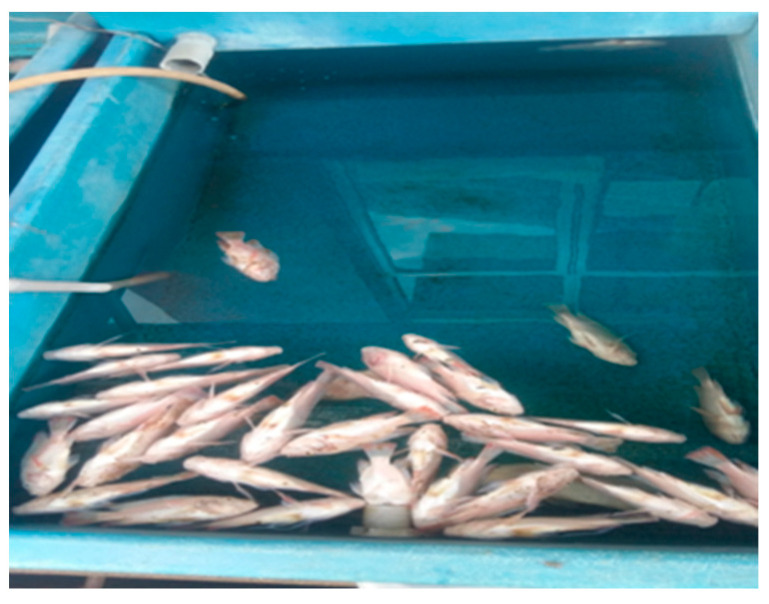
All Red hybrid tilapia died in control Group Cx after the challenged at 4-wpv.

**Figure 6 vaccines-09-00051-f006:**
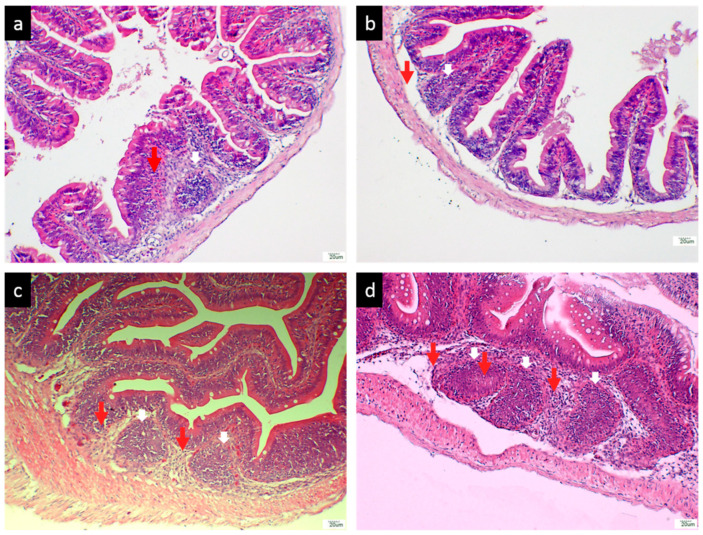
Histological sections of the hindgut of Red hybrid tilapia fed the formalin inactivated *S. iniae* vaccine (FKV) (**a**) Group 1A, tilapia fed with FKV vaccines for 3 days, developed the structure of GALT (red arrow) in the lamina propria, haematoxylin and eosin (H & E) × 200. (**b**) Group 1B, fed with FKV vaccines for 6 days, developed the structure of GALT (red arrow) in the lamina propria, H & E × 200. (**c**) Group 2A, fed with FKV vaccines for 9 days, developed two structures of GALTs (red arrows) in the lamina propria, H & E × 200. (**d**) Group 2B, fed with FKV vaccines for 9 days (2 times booster dose), developed more structures of GALTs (red arrows) in the lamina propria, H & E × 200.

**Figure 7 vaccines-09-00051-f007:**
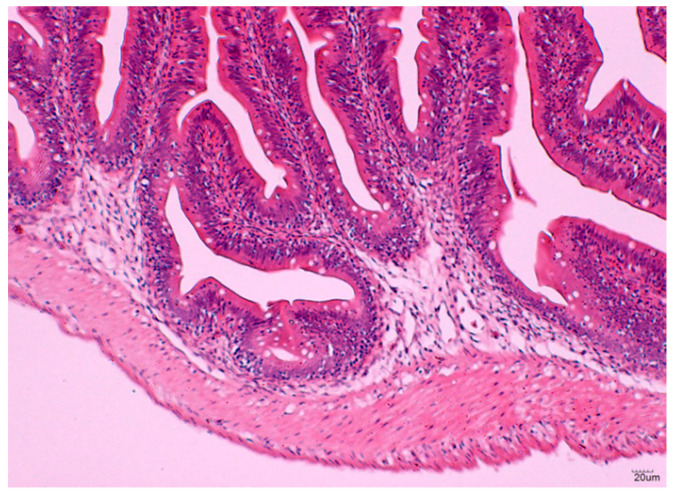
Group Cx, fed with normal feed Cargill, no structure of GALT developed in the lamina propria, H & E × 200.

**Figure 8 vaccines-09-00051-f008:**
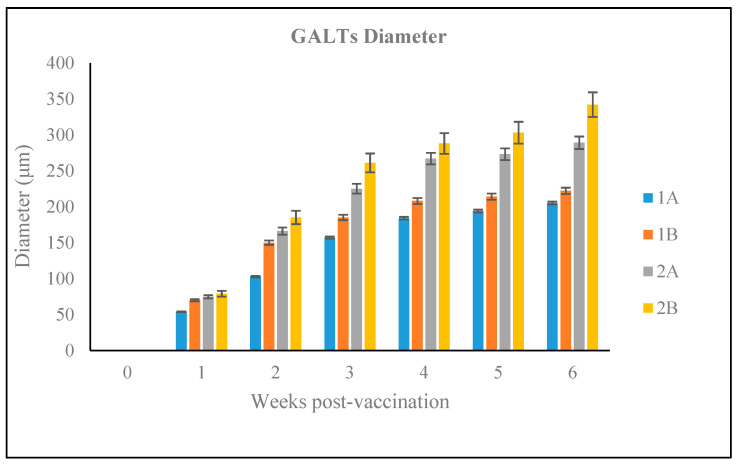
GALTs diameter observed in experimental groups after oral vaccination in the 6-week trial.

**Figure 9 vaccines-09-00051-f009:**
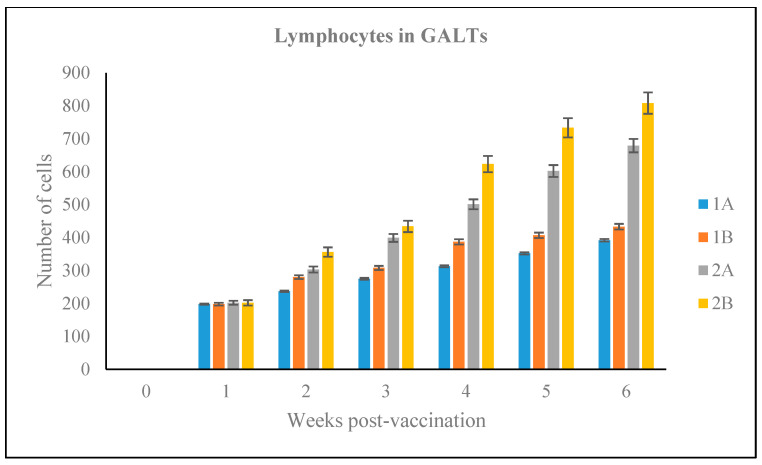
Estimated number of lymphocytes cells in GALT observed in experimental groups during the 6-week vaccination trial.

**Table 1 vaccines-09-00051-t001:** Design of feeding trial (treatment groups, vaccine feeding period) in the Red hybrid tilapia.

Groups	Vaccine Feeding Rate and Amount	Vaccine (FKV) Feeding Days	No. of Fish
1A	2 times/day 50 g	3	60
1B	2 times/day 50 g	6	60
2A	2 times/day 50 g	9	60
2B	2 times/day 50 g	9 (Booster on day 14 and 21)	60
Cx	2 times/day Normal feed 50 g	no vaccine	60

**Table 2 vaccines-09-00051-t002:** The number, percentage survival and percentage mortality in Red hybrid tilapia after challenged with 1 × 10^6^ CFU/mL live *S. iniae* by intraperitoneal injection.

Groups	Number of Fish	Mortality Days Post-Challenge														Survival (%)	Mortality (%)
		1	2	3	4	5	6	7	8	9	10	11	12	13	14		
1A	40	10	6	2	4	2	-	2	2	1	-	1	-	2	-	20	80
1B	40	8	2	4	2	1	1	-	1	1	-	1	1	-	-	45	55
2A	40	6	2	-	2	-	2	2	2	1	2	-	-	-	-	55	45
2B	40	2	-	2	1	2	-	2	1	1	1	-	-	-	-	70	30
Cx	40	16	8	4	6	2	2	2	-	-	-	-	-	-	-	0	100

**Table 3 vaccines-09-00051-t003:** Isolation of *S. iniae* from various organs of vaccinated fish (Groups 1A, 1B, 2A, 2B, and control Group Cx at different time intervals post-challenge.

Group	hpc	Brain	Eye	Kidney	Group	hpc	Brain	Eye	Kidney	Group	hpc	Brain	Eye	Kidney
1A	6	+	+	-	2A	6	+	-	-	Cx	6	+	+	+
1B	16	+	+	+	2B	16	-	-	+	Cx	16	+	+	+
1A	24	+	+	-	2A	24	-	-	-	Cx	24	+	+	+
1B	32	+	+	-	2B	32	-	-	-	Cx	32	+	+	+
1A	40	+	-	+	2A	40	-	-	-	Cx	40	+	+	+
1B	48	-	-	+	2B	48	-	-	-	Cx	48	+	+	+

hpc: hours post-challenge.

## Data Availability

Data sharing not applicable. No new data were created or analyzed in this study. Data sharing is not applicable to this article.

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
