# Peer review of "Efficacy of Feed-Based Formalin-Killed Vaccine of Streptococcus iniae Stimulates the Gut-Associated Lymphoid Tissues and Immune Response of Red Hybrid Tilapia"

_vaccines, 2021, doi:10.3390/vaccines9010051_

Round 1

Reviewer 1 Report

This is a revised version of a previously submitted manuscript. During revision, the manuscript has improved a lot. Most of the points from my evaluation report on the previous version of this manuscript are addressed to a sufficient extent. In addition, the language has improved significantly. In the present version, I spotted a few errors, which slipped through the authors' final editing, and which are mared in the annotated manuscript attached to my evaluation report. I have no further requests for changes.

Author Response

We followed the changes according to the reviewer 1 

Reviewer 2 Report

The authors did not really change much in response to my comments.

Author Response

Thanks. 

Reviewer 3 Report

Appropiate for publication

Author Response

Thanks. 

This manuscript is a resubmission of an earlier submission. The following is a list of the peer review reports and author responses from that submission.

Round 1

Reviewer 1 Report

This study describes vaccination experiments in Tilapia against an infection with Streptococcus iniae, which causes a septicaemic infection in these fish and is responsible for high losses. The development of a vaccine for protection against S. iniae was addressed in several previous studies, however in the present manuscript, the authors report their studies on the application of an oral vaccine. Therefore, the study is of interest and addresses a novel approach.

For a vaccine candidate, the authors use formalin killed bacteria, which were included into the feed and fed to the tilapia for different periods of time. In addition, some experimental groups received a second and a third dosis after 2 and 3 weeks post initial start of the feeding. A challenge infection was done by intra peritoneal injection of virulent bacteria into fish from all vaccinated groups by 4 weeks post initial feeding of the vaccine candidate. Fish from the group fed until week 3 survived the infection at a ratio of 70 % of infected fish, while fish which were fed only for a short period of time experience mortality at a ratio of 80 %.

There is one main point of concern with this study: The challenge infection was performed at a time, when fish of the groups with high survival still had a high antibody titre in blood. In this respect, it cannot be excluded, that the antibody from the vaccine was limiting bacterial growth rather than the protective effect of the vaccine. This limits the value of the study.

Other, minor points: Even though the experiment was done in the way that fish from the several vaccine groups were distributed over several aquaria, the mortality data are presented for the entire groups (feeding short period of time, feeding for a prolonged period, booster-feedings). This does not allow to see a variation in the survival time among different vaccinated groups and a statistical analysis of the survival of the fish (e.g. by Kaplan Meyer evaluation or log rank evaluation) could not be done.

In the graphs displaying the antibody titres in serum, mucus and intestinal lavage, significant differences to the control should be indicated.

In Materials and methods, it is not described, how the size of the GALT was determined and the leukocytes were calculated.

In addition, the manuscript needs serious language editing.

This study describes vaccination experiments in Tilapia against an infection with Streptococcus iniae, which causes a septicaemic infection in these fish and is responsible for high losses. The development of a vaccine for protection against S. iniae was addressed in several previous studies, however in the present manuscript, the authors report their studies on the application of an oral vaccine. Therefore, the study is of interest and addresses a novel approach.

For a vaccine candidate, the authors use formalin killed bacteria, which were included into the feed and fed to the tilapia for different periods of time. In addition, some experimental groups received a second and a third dosis after 2 and 3 weeks post initial start of the feeding. A challenge infection was done by intra peritoneal injection of virulent bacteria into fish from all vaccinated groups by 4 weeks post initial feeding of the vaccine candidate. Fish from the group fed until week 3 survived the infection at a ratio of 70 % of infected fish, while fish which were fed only for a short period of time experience mortality at a ratio of 80 %.

There is one main point of concern with this study: The challenge infection was performed at a time, when fish of the groups with high survival still had a high antibody titre in blood. In this respect, it cannot be excluded, that the antibody from the vaccine was limiting bacterial growth rather than the protective effect of the vaccine. This limits the value of the study.

Other, minor points: Even though the experiment was done in the way that fish from the several vaccine groups were distributed over several aquaria, the mortality data are presented for the entire groups (feeding short period of time, feeding for a prolonged period, booster-feedings). This does not allow to see a variation in the survival time among different vaccinated groups and a statistical analysis of the survival of the fish (e.g. by Kaplan Meyer evaluation or log rank evaluation) could not be done.

In the graphs displaying the antibody titres in serum, mucus and intestinal lavage, significant differences to the control should be indicated.

In Materials and methods, it is not described, how the size of the GALT was determined and the leukocytes were calculated.

In addition, the manuscript needs serious language editing.  

Author Response

This study describes vaccination experiments in Tilapia against an infection with Streptococcus iniae, which causes a septicaemic infection in these fish and is responsible for high losses. The development of a vaccine for protection against S. iniae was addressed in several previous studies, however in the present manuscript, the authors report their studies on the application of an oral vaccine. Therefore, the study is of interest and addresses a novel approach.

For a vaccine candidate, the authors use formalin killed bacteria, which were included into the feed and fed to the tilapia for different periods of time. In addition, some experimental groups received a second and a third dosis after 2 and 3 weeks post initial start of the feeding. A challenge infection was done by intra peritoneal injection of virulent bacteria into fish from all vaccinated groups by 4 weeks post initial feeding of the vaccine candidate. Fish from the group fed until week 3 survived the infection at a ratio of 70 % of infected fish, while fish which were fed only for a short period of time experience mortality at a ratio of 80 %.

There is one main point of concern with this study: The challenge infection was performed at a time, when fish of the groups with high survival still had a high antibody titre in blood. In this respect, it cannot be excluded, that the antibody from the vaccine was limiting bacterial growth rather than the protective effect of the vaccine. This limits the value of the study.

Other, minor points: Even though the experiment was done in the way that fish from the several vaccine groups were distributed over several aquaria, the mortality data are presented for the entire groups (feeding short period of time, feeding for a prolonged period, booster-feedings). This does not allow to see a variation in the survival time among different vaccinated groups and a statistical analysis of the survival of the fish (e.g. by Kaplan Meyer evaluation or log rank evaluation) could not be done.

In the graphs displaying the antibody titres in serum, mucus and intestinal lavage, significant differences to the control should be indicated.

  • We are preferred not to indicates significant icons in these graphs (unless we change to bar graphs then, it would be nice to put the significant icons. We had several publications without significant different in the graph.

In Materials and methods, it is not described, how the size of the GALT was determined and the leukocytes were calculated.

  • We are using Image analyser by Olympus to measure the GALT and counting the MNCs within the GALT.

In addition, the manuscript needs serious language editing.

  • The manuscript has been edited by Dr Kim D Thompson from UK.

Reviewer 2 Report

Why not IgT? Even by PCR? Tilapia have been reported to have IgT, so measuring immunoglobulins in mucous should include measuring them. When looking at GALT lymphocytes, could they be stained for IgM to show what types of lymphocytes they are.

Are antibodies protective for this bacteria - more detail could be provided in the introduction, with some discussion of other mechanisms.

Why only boiling for the bacteria that is used to coat the plate? This does not make for efficient coating as the bacteria are not efficiently lysed – they should be homogenized or bead beaten. Was a BCA assay used on the lysate to make sure there was sufficient protein for coating? Was this done before each assay to ensure the same amount of protein was used each time – otherwise the coating amount might be inconsistent.

Figure 6: It is really difficult to see the white arrows right away – can they be changed to a colour that is more visibe.

Table 3: the wrap around of text makes it difficult to read. It should be fixed.

PCR is not a definitive test for the presence of viable bacteria – isolating it and culturing would be better.

Author Response

Why not IgT? Even by PCR? Tilapia have been reported to have IgT, so measuring immunoglobulins in mucous should include measuring them. When looking at GALT lymphocytes, could they be stained for IgM to show what types of lymphocytes they are.

  • IgT is consider new findings and none of the commercially available for anti-IgT HRP available in the market compared to anti-IgM HRP. Therefore, is not easy to run IgT-based ELISA and of course, we love to run ELISA focus on IgT instead of IgM to add value to our findings.  
  • Types of lymphocytes are required the suitable biomarkers and I guess it can be done but expensive to buy those several biomarkers.

Are antibodies protective for this bacteria - more detail could be provided in the introduction, with some discussion of other mechanisms.

  • To me this protection by antibodies can be discuss in review paper in details and our discussion is already too long to discuss other mechanisms.

Why only boiling for the bacteria that is used to coat the plate? This does not make for efficient coating as the bacteria are not efficiently lysed – they should be homogenized or bead beaten. Was a BCA assay used on the lysate to make sure there was sufficient protein for coating? Was this done before each assay to ensure the same amount of protein was used each time – otherwise the coating amount might be inconsistent.

  • This ELISA technique has been established since we are studied pneumonic mannheimiosis in small ruminants and then we applied the technique in fish immune response resulting in consistent results. Even the cut-off was also run for at least 50 normal fish.

Figure 6: It is really difficult to see the white arrows right away – can they be changed to a colour that is more visibe.

  • We changed to red arrows.

 Table 3: the wrap around of text makes it difficult to read. It should be fixed.

  • We adjusted it accordingly.

 PCR is not a definitive test for the presence of viable bacteria – isolating it and culturing would be better.

  • We did bacterial isolation before run PCR; PCR is specifically to confirm the isolates is S. iniae.

Reviewer 3 Report

The manuscript with code vaccine-8361 demonstrated that an oral vaccine formed by formaline-killed bacteria and adsorbed onto normal dry-diet is able to induce the formation of IgM specific antibodies. The authors used different administration times and also compared one vaccination with other procedures with two boosts and found that the vaccine is able to increase the amount of IgM in several tissues including mucus tissue. The methods are accurate and the conclusions are supported by the data.

Minor comments:

Page 2 lien 78: the dietary ratio should be 2% of the tank biomase instead of body weight.

Line 152, The unit abbreviation of microlitres should be corrected.

Material and method point 2.6.7. The authors have to include how they quantify the diameter of GALT structure and the number of lymphocyte on them. How many sections did they used form each fish? Where those sections obtained randomly through the organ or tissue fraction they collected previously? How many fish per group did they analyze?

Statisticts: Why the authors did not used letters to show statistically differences in their graph? It would help to understand the differences of the different treated group easily.

Results:

It would be easier to analyze the data about the IgM levels knowing first what had happened with the infection trial.

Point 3.4.1. All this had already been described in point 3.3. of the results. It is not needed to write it again.

Discussion:

Authors should include an evaluation of the management of the feeding with the vaccine for economical purposes. Will be possible to have a diet with the vaccine store for long time? Would be necessary to adjust the vaccine after the first used? In that case, will be possible to do it in the aquaculture enterprises?

Conclusions are accurate to the data; Authors should define that they have only analyzed the humoral specific immune response upon vaccination. Moreover, the manuscript did not dealt with long term protection of the vaccine. How much time will the protection last? Obviously, author cannot concluded nothing about this issue as they have only 6 weeks of sampling, but probably they could included something about it in the discussion section.

Author Response

The manuscript with code vaccine-8361 demonstrated that an oral vaccine formed by formaline-killed bacteria and adsorbed onto normal dry-diet is able to induce the formation of IgM specific antibodies. The authors used different administration times and also compared one vaccination with other procedures with two boosts and found that the vaccine is able to increase the amount of IgM in several tissues including mucus tissue. The methods are accurate and the conclusions are supported by the data.

Minor comments:

Page 2 lien 78: the dietary ratio should be 2% of the tank biomase instead of body weight.

  • We changed it accordingly.

Line 152, The unit abbreviation of microlitres should be corrected.

  • We changed it accordingly.

Material and method point 2.6.7. The authors have to include how they quantify the diameter of GALT structure and the number of lymphocyte on them. How many sections did they used form each fish? Where those sections obtained randomly through the organ or tissue fraction they collected previously? How many fish per group did they analyze?

  • We have answered it in Reviewer 1 and in the M&M.

Statisticts: Why the authors did not used letters to show statistically differences in their graph? It would help to understand the differences of the different treated group easily.

  • We have answered it in Reviewer 1

Results:

It would be easier to analyze the data about the IgM levels knowing first what had happened with the infection trial.

  • Why should you to analyze IgM in the infection trial… you will not have enough data as the fish will die quickly due to S. iniae nature that causing a septicaemic condition in fish, even the antibody will not develop as soon the infection is progress acutely.

Point 3.4.1. All this had already been described in point 3.3. of the results. It is not needed to write it again.

Discussion:

Authors should include an evaluation of the management of the feeding with the vaccine for economical purposes. Will be possible to have a diet with the vaccine store for long time? Would be necessary to adjust the vaccine after the first used? In that case, will be possible to do it in the aquaculture enterprises?

  • These suggestions should be in the review paper as they can discuss these matters in detail.

Conclusions are accurate to the data; Authors should define that they have only analyzed the humoral specific immune response upon vaccination. Moreover, the manuscript did not dealt with long term protection of the vaccine. How much time will the protection last? Obviously, author cannot concluded nothing about this issue as they have only 6 weeks of sampling, but probably they could included something about it in the discussion section.

  • You should remember that the vaccine development based on how long the fish rear on the farm when the disease to be likely to occur on the farm and in what season to cause outbreaks.
  • In Malaysia, tilapia farming is only for 6 months before harvesting, so the farms are not required for a long-time immunity. At least, the fish have some protection for a short period of time just before harvesting the fish.